

# Plant height variation and genetic diversity between *Prunus ledebouriana* (Schlecht.) YY Yao and *Prunus tenella* Batsch based on using SSR markers in East Kazakhstan

Aidyn Orazov[1,2,3,4], Moldir Yermagambetova[2,3], Anar Myrzagaliyeva[1], Nashtay Mukhitdinov[3], Shynar Tustubayeva[1], Yerlan Turuspekov[2,3] and Shyryn Almerekova[2,3]

[1] School of Natural Sciences, Astana International University, Astana, Kazakhstan
[2] Molecular Genetics Laboratory, Institute of Plant Biology and Biotechnology, Almaty, Kazakhstan
[3] Faculty of Biology and Biotechnology, Al-Farabi Kazakh National University, Almaty, Kazakhstan
[4] Faculty of Biology, University of Warsaw, Warsaw, Poland

Corresponding authors
Shynar Tustubayeva,
tustubaeva.shynar@gmail.com
Shyryn Almerekova,
almerekovakz@gmail.com

## ABSTRACT

**Background**. Genetic differences between isolated endemic populations of plant species and those with widely known twin species are relevant for conserving the biological diversity of our planet's flora. *Prunus ledebouriana* (Schlecht.) YY Yao is an endangered and endemic species of shrub almond from central Asia. Few studies have explored this species, which is closely related and morphologically similar to the well-known *Prunus tenella* Batsch. In this article, we present a comparative analysis of studies of three *P. ledebouriana* populations and one close population of *P. tenella* in Eastern Kazakhstan in order to determine the particular geographic mutual replacement of the two species.

**Methods**. The populations were collected from different ecological niches, including one steppe population near Ust-Kamenogorsk (*P. tenella*) and three populations (*P. ledebouriana*) in the mountainous area. Estimation of plant height using a $t$-test suggested a statistically significant difference between the populations and the two species ($P < 0.0001$). DNA simple sequence repeat (SSR) markers were applied to study the two species' genetic diversity and population structure.

**Results**. A total of 19 polymorphic SSR loci were analyzed, and the results showed that the population collected in mountainous areas had a lower variation level than steppe populations. The highest level of Nei's genetic diversity index was demonstrated in the 4-UK population (0.622) of *P. tenella*. The lowest was recorded in population 3-KA (0.461) of *P. ledebouriana*, collected at the highest altitude of the four populations (2,086 meters above sea level). The total genetic variation of *P. ledebouriana* was distributed 73% within populations and 27% between populations. STRUCTURE results showed that two morphologically similar species diverged starting at step $K = 3$, with limited population mixing. The results confirmed the morphological and genetic differences between *P. tenella* and *P. ledebouriana* and described the level of genetic variation for *P. ledebouriana*. The study's results proved that the steppe zone and mountain altitude factor between *P. tenella* and isolated mountain samples of *P. ledebouriana*.

## INTRODUCTION

The systematization of taxa of endemic wild mountain plant species is an urgent issue in the latest taxonomy of the Rosaceae Juss family; one of the most prominent examples is the systematization of representatives of endemic mountain shrubs of the almond subgenus. The almond is one of the most essential cultivated and wild plant species worldwide. Shrub forms are often used in introductory and subsequent material for landscaping large cities in Kazakhstan. Studying isolated endemic populations will allow for the comparison of genetic variation among the genus' general distribution area species, as some medicinal properties of almonds are also known. Specimens from pristine mountain populations are potential carriers of valuable biological and chemical compounds (*Gradziel, 2011*). Almond plants are members of the genus *Prunus* L. (tribe *Amygdaleae*), one of 65 genera of the subfamily *Prunoideae* of the complex family Rosaceae Juss. The genus is represented by four subgenera (subgen. *Amygdalus* (L.) Focke., *Cerasus* (Mill.) A.Gray., *Emplectocladus* (Torr.) A.Gray., and *Prunus* L.) and includes about 254 species. The subgenus *Amygdalus* consists of six sections: *Amygdalopsis* (Carr.) Linsz., *Cerasioides* (Carr.) Linsz., *Chamaeamygdalus* Spach, *Euamygdalus* Spach., *Lycioides* Spach., and *Spartioides* Spach. One of the least studied sections is the pygmy almond *Chamaeamygdalus*, which has low yield, a special protection status (endemic and rare plant species), and ornamental properties (*Artemov et al., 2009*; *Browicz & Zohary, 1996*). According to the list of vascular plants in Kazakhstan (*Abdulina, 1999*; *Komorov, 1941*), there are three species in the flora of Kazakhstan: *Chamaeamygdalus*—steppe almond (*Prunus tenella* Batsch syn. *Amygdalus nana* L.), Ledebour's almond (*Prunus ledebouriana* (Schlecht.) YY Yao syn. *Amygdalus ledebouriana* (*Schlechtendal, 1854*)), and Pettunnikov's almond (*Prunus petunnikowii* (Litv.) Rehder syn. *Amygdalus petunnikowii* Litw.). In addition, the list of flora of Kazakhstan includes one species of the section *Lycioides* (*Amygdalus communis* L. syn. *Prunus dulcis* (Mill.) D.A. Webb.) and one species of the section *Euamygdalus* (*Amygdalus spinosissima* Bunge. syn. *Prunus spinosissima* (Bunge) Franch.). *P. tenella* is widespread in Southern Europe and the European part of Asia, mainly in the steppe zones, and is available for cultivation. According to the flora of the Kazakh Soviet Socialist Republic (Kazakh SSR), *P. ledebouriana* is an endemic species for Kazakhstan, growing in the Altai, Tarbagatai, and Dzungarian Alatau mountains and replacing *P. tenella* in east Kazakhstan (*Pavlov, 1961*). *P. ledebouriana* is listed in the Red Book of the Republic of Kazakhstan (*Baitulin, 2014*) and the Book of Woody Plants of Central Asia (*Eastwood, Lazkov & Newton, 2009*).

Various international databases have interpreted the status of *P. ledebouriana* (*The Plant List* (TPL) and *World Flora Online* (WFO)) and attempted to determine the taxonomy of this species (*GBIF, 2020*). *P. ledebouriana* is closely related to the steppe almond *P. tenella*, as they have similar morphological features and are distinguished by a relatively large habitus (plant height), sizes of leaves, and fruits (*Orazov et al., 2020*). *P. tenella* is

the southernmost species of section *Chamaeamygdalus*, has a wide distribution from the northern Balkans to Kazakhstan and China (*Ladizinsky, 1999*), and is often used for cultivation as an ornamental species. These two related species have no boundaries and the genetic differences are poorly understood.

According to a report on the flora of China, *P. tenella* has a synonymous name, *A. nana* (syn.) (*Lu et al., 2003*). Across various literary sources, two synonymous species have been recorded in the territory of East Kazakhstan (an administrative region of Kazakhstan bordering Russia and China), adding complexity to identifying the species. The distribution of *P. tenella* among studied territories is not uniform since the area consists of several isolated populations. According to various sources, *P. tenella* predominates in the steppes and low hills of the low mountains of the Kalba and Ulba ranges in the Altai Mountains (*Planetarium*).

*P. ledebouriana* is found in the cold and xerophytic mountain areas adjacent to Russia (Narym Range of the Altai Mountains) and China (foothills of the Tarbagatai Range) (*Orazov et al., 2020*). The population in the foothills of Tarbagatai is the most extensive and is included in the Red Book of Kazakhstan (*Stepanova, 1962*; *Baitulin, 2014*). Natural populations of *P. ledebouriana* are declining due to habitat degradation, frequent droughts, changes in the fire regime (succession), overgrazing, and urbanization (*Sumbembaev, 2018*; *Sumbembayev et al., 2021*; *Aidarkhanova et al., 2022*; *Kusmangazinov et al., 2023*). Therefore, by the Decree of the Government of the Republic of Kazakhstan in 2018, the Tarbagatai State National Natural Park (East Kazakhstan region, Urdzhar district) was adopted and the preservation of this rare and endangered species was recommended (*Republic of Kazakhstan, 2018*). The morphological similarity of the two species complicates the protection of this endemic plant species (*Potter et al., 2002*). In this regard, making a clear distinction between the two species of wild almond populations in Eastern Kazakhstan is crucial.

*P. ledebouriana* primarily reproduces vegetatively in addition to sexual reproduction. It usually flowers from April to May and bears fruit from June to July (*Pavlov, 1961*; *Bin et al., 2008*; *Orazov et al., 2022*). Isolating factors include the presence of Lake Zaisan and the Irtysh River that feeds it; the location of the Zaisan basin between Altai and Tarbagatai and the non-contiguous Ulba, Kalba, Narym ridges of the Altai mountains; and the remote location of the Tarbagatai ridge of the Saur-Tarbagatai mountains (*Egorina, Zinchenko & Zinchenko, 2003*).

Applying molecular methods in botany and plant systematics has provided opportunities for identifying and confirming species and their taxonomic position in the genus (*Andersen & Lübberstedt, 2003*). Various types of DNA markers have been successfully used to assess the genetic diversity of species of *Prunus*. These studies included the use of random amplified polymorphism of DNA (RAPD) (*Casas et al., 1999*), inter simple simples sequence repeats (ISSR) (*Martins, Tenreiro & Oliveira, 2003*), amplified fragment length polymorphism (AFLP) (*Struss et al., 2003*), and simple sequence repeats (SSR) (*Aranzana et al., 2003*) markers. One of the most informative types of DNA markers are microsatellite markers (SSR), which are characterized by a high level of polymorphism and codominant inheritance (*Kalendar, 2011*; *Genievskaya et al., 2020*). Analysis of population genetics

using microsatellite markers provides information on the overall levels of genetic diversity, genetic structure, and effective population size, which are typically critical when developing effective management strategies for the research of genetic resources of endemic plant species (*Turuspekov & Abugalieva, 2015*; *Abugalieva & Turuspekov, 2017*; *Almerekova et al., 2018*; *Almerekova et al., 2020*; *Genievskaya et al., 2020*). Various microsatellite markers have been successfully used to study the phylogenetic relationships between various cultivated almonds (common almond *P. dulcis* (Mill.) DA Webb.) and their wild relatives (*Xu et al., 2004*; *Xie et al., 2006*; *Shiran et al., 2007*; *Sorkheh et al., 2007*; *Zhang et al., 2018*; *Zargar et al., 2023*). However, there have been limited studies on the genetic analysis of natural populations of wild species in the subgenus *Amygdalus* (*Varshney, Graner & Sorrells, 2005*; *Tahan et al., 2009*). This study is one of the first to explore the mountain populations of *P. ledebouriana* and obtain genetic information. The application of SSR markers in population genetic analysis for the narrowly endemic species *P. ledebouriana* can be successfully used to study the genetic diversity and population structure of the natural population in Eastern Kazakhstan.

## MATERIALS & METHODS

### Study of genetic structure using SSR

This study investigated three isolated populations of *P. ledebouriana* from two mountain geographic ranges (Altai and Tarbagatai) and one *P. tenella* from Eastern Kazakhstan. Among these four populations, the first one (1-UR) was collected on several isolated gorges in the state national natural park ''Tarbagatai'' at the height of the shrub belt. The materials of the second and third populations were collected in the cold and xerophytic mountain regions of the Kalba (2-KO) and Narym (3-KA) ridges. The population of *P. tenella* (4-UK) was collected from a small hilly plain (steppe) zone in the outlines of the Kalbinskiy and Ulbinskiy ridges in the border zone of the city of Ust-Kamenogorsk and the village of Novo-Akhmirovo (Table S1). The plant height of *P. ledebouriana* was measured according to *Goloskokov (1972)*. In total, 20 leaves from each of the 60 *P. ledebouriana* and 20 *P. tenella* plant populations were collected. The distances between populations were at least 100 kilometers, and plants within a population were selected at a distance of at least 50 m from each other.

Fresh plant leaves were used for DNA extraction. Total DNA was isolated from crushed leaf powder according to the Cetyl trimethyl ammonium bromide (CTAB) protocol with double purification with chloroform (*Doyle & Doyle, 1987*). The quality and concentration of DNA were assessed using a NanoDrop 2000 spectrophotometer (Thermo Fisher Scientific, Waltham, MA, USA) and electrophoresis in 1% agarose gel. DNA concentration was normalized to the working concentration for further analysis.

Twenty-two SSR markers of the nuclear genome were used as DNA markers and selected according to *Mnejja et al. (2005)*. PCR amplification was performed in 10 ul reaction volume containing 20 ng template DNA, one PCR buffer, 1.5 mM of MgCl2, 0.2 mM of dNTPs, 0.4 mM of each primer, and one unit of Taq DNA polymerase (Sileks, Badenweiler, Germany). PCR conditions were set at 95 °C for 1 min, followed by 35 cycles

of 94 °C for 30 s, 50–65 °C for 30 s, 72 °C for 30 s, and 5 min at 72 °C for final elongation. PCR products were separated on a 6% polyacrylamide gel using 0.5x Tris-borate-EDTA buffer. DNA fragments were identified using an ethidium bromide staining procedure. Alleles were determined using 100-bp ladders (Thermo Fisher Scientific). Visualization was performed using the GelDoc XR+ gel documentation system (Bio-Rad, Hercules, CA, USA).

### Statistical analysis and polymorphism information content

Descriptive statistics and t-tests (SPSS Statistics v.27.0; https://www.ibm.com/products/spss-statistics) were used to describe morphological features and identify population differences (plant height across different populations of *P. ledebouriana* and *P. tenella*).

Each DNA fragment obtained was treated as a separate character and evaluated as a discrete variable. Accordingly, rectangular binary data matrices were obtained for SSR markers.

To assess the effectiveness of markers, the following polymorphism indices were used: marker index (MI), resolving power (Rp), observed heterozygosity (HO), expected heterozygosity (HE), polymorphism information content (PIC values), inbreeding coefficient (FIS), and fixation index, (Fst) as estimated by *Weir & Cockerham (1984)*.

The genetic diversity of *P. ledebouriana*, including the Shannon diversity index (I), was assessed using PopGene (*Yeh et al., 2000*). Wright's F-statistics was calculated for each SSR locus using the GenAlEx 6.5 program (*Peakall & Smouse, 2006*). Analysis of molecular variance (AMOVA) was performed across organizations using GenAlEx 6.5 (*Peakall & Smouse, 2006*). Principal coordinate analysis (PCA) was performed using the Numerical Taxonomy and Multivariate Analysis System (NTSYS-pc) program (*Rohlf, 1998*). The STRUCTURE program also applied Bayesian cluster analysis (*Pritchard, Stephens & Donnelly, 2000*). The dendrogram was constructed using the PAST program and unweighted pair group method with arithmetic mean (UPGMA) algorithm and Boot N:1,000 (*Joshi et al., 2000*). Cluster analysis of *P. ledebouriana* and *P. tenella* was studied using STRUCTURE. An admixture model was used, which made it possible to analyze the frequencies of admixtures and correlated alleles. Five independent simulations were run, each including 100,000 burn-in steps and a subsequent 100,000 Markov chain Monte Carlo (MCMC) iterations.

## RESULTS

### Variability of plant height among populations

The samples of three *P. ledebouriana* and one population of *P. tenella* were assessed according to plant height (Fig. 1). The most significant plant height was recorded for the 3-KA population (2.09 ± 0.06 m), which was the mountain population's highest elevation above sea level (Table S1). Similar parameters for plant height were recorded in representatives from two other mountain populations: 2-KO and 1-UR, 1.79 ± 0.03 m and 1.78 ± 0.03 m, respectively. The lowest height values of plants in the steppe populations of *P. tenella* were recorded in 4-UK (1.41 ± 1.04 m). The *T*-test confirmed a statistically significant difference between the plant height of the three populations of *P.*

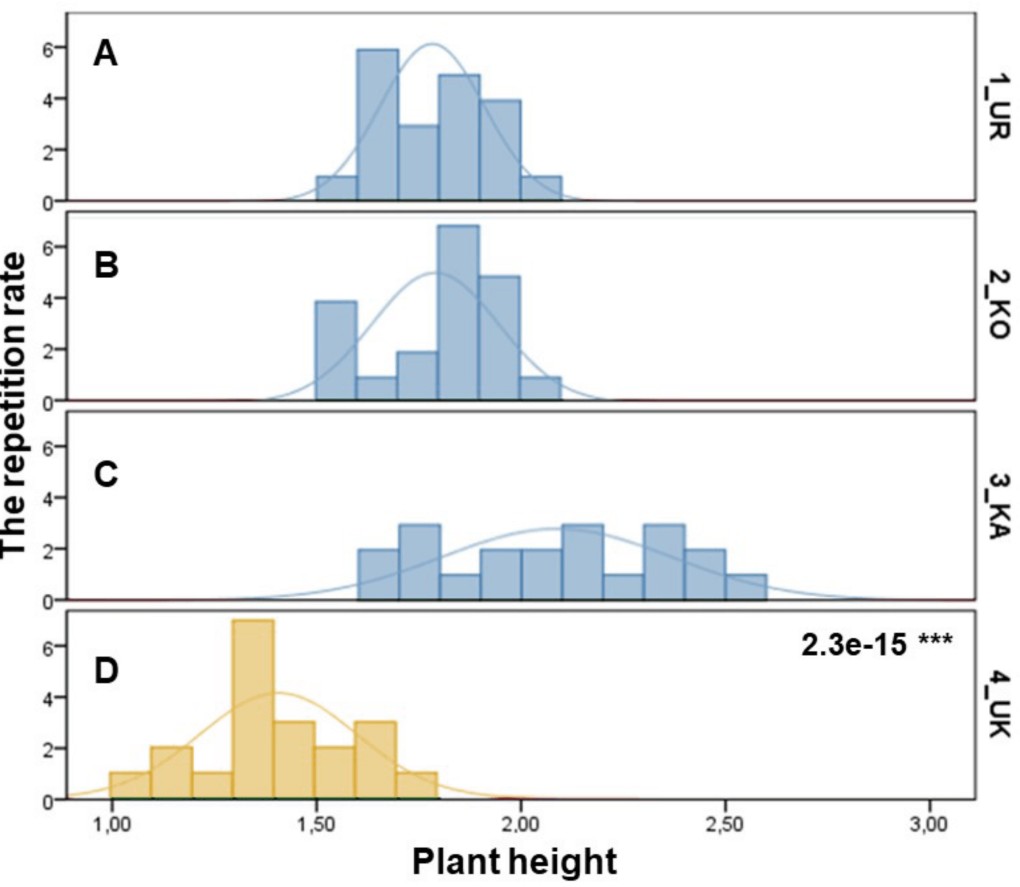

**Figure 1** Column diagram of plant heights in populations of *P. ledebouriana* (A—Urdzhar, B—Kokpekty, C—Katon-Karagai-blue) and *P. tenella* (D—Ulansky-yellow).

*ledebouriana*: 3-KA, 2-KO, and 1-UR (*P* value = 2.3e−15). The *t*-test suggested that all three mountainous populations had significantly higher plants than the steppe population *P. tenella* 4-UK (*P* < 0.0001).

## Allelic variation of SSRs

The assessment of allelic variations in *P. ledebouriana* and *P. tenella* showed that 19 out of the studied 22 markers were polymorphic (Table S2). Figure 2 shows two gel snapshots of different populations with high polymorphism. It appeared that CPDCT038 had three loci. Information on these 19 polymorphic SSR loci, including the sizes of bands, is presented in Table 1.

Results showed that the three SSR loci had two alleles, 11 SSR loci had three, three SSR markers had four, and two loci had five. The results of genotyping using 19 SSR loci and their statistical details are shown in Table 2. It was determined that 19 loci, according to the polymorphism level (PIC values) could be separated into three subpopulations. The first group was comprised of five SSR loci with a PIC above 0.5, the second group of eight

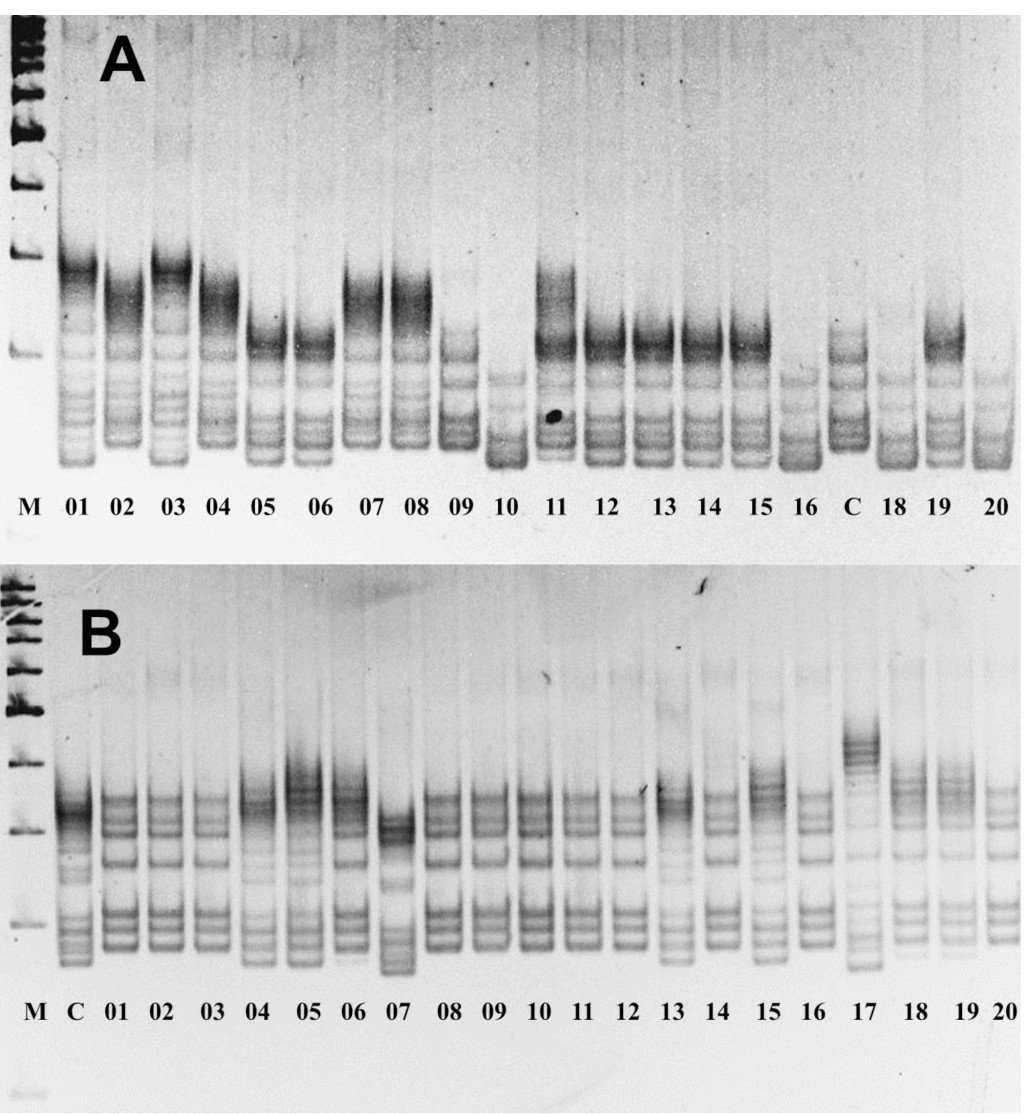

**Figure 2** **Gel images of the results of electrophoresis of two polymorphic SSR loci, (A) *P. ledebouriana* (CPDCT045) and (B) *P. tenella* (CPDCT025).**

loci with a PIC between 0.3 and 0.5, and the third group of the remaining six loci with a PIC below 0.3. The mean PIC value was 0.38 (Table 3).

The genetic diversity assessment using Nei's index suggested that the highest genetic diversity was in *P. tenella* 4-UK (0.622), followed by three populations of *P. ledebouriana*: 1-UR, 2-KO, and 3-KA (Table 2). The average genetic diversity index for the three populations of *P. ledebouriana* was 0.501. To compare inter-population diversity, two main scenarios were used. The first included the assumption of the unity of four populations as representatives of the species *P. ledebouriana*, and the second scenario assumed the presence of three populations of *P. ledebouriana* and one population of *P. tenella*. The AMOVA suggested that the total genetic diversity in *P. ledebouriana* could be partitioned as

**Table 1** The list of polymorphic SSR loci selected from *Mnejja et al. (2005)* and analyzed in plants of *P. ledebouriana* and *P. tenella*.

| # | Loci | Number of alleles | Motif | Annealing temperature optimized (°C) | Size obtained (bp) |
|---|---|---|---|---|---|
| 1. | CPDCT005 | 3 | $(CT)_{14}$ | $64^+$ | 96, 98, 101 |
| 2. | CPDCT007 | 3 | $(GA)_{19}$ | $48^+$ | 163, 172, 183 |
| 3. | CPDCT008 | 3 | $(GA)_{18}$ | 62 | 185, 188, 192 |
| 4. | CPDCT012 | 2 | $(GA)_{12}$ | $56^+$ | 157, 166 |
| 5. | CPDCT015 | 4 | $(CT)_{20}$ | 62 | 214, 215, 216, 217 |
| 6. | CPDCT016 | 3 | $(GA)_{19}$ | 62 | 173, 178, 193 |
| 7. | CPDCT022 | 3 | $(CT)_{17}$ | 62 | 152, 158, 159 |
| 8. | CPDCT023 | 3 | $(GA)_{9}$ | 62 | 171, 176, 179 |
| 9. | CPDCT025 | 4 | $(CT)_{10}$ | 62 | 184, 188, 194, 195 |
| 10. | CPDCT027 | 3 | $(CT)_{19}$ | 62 | 174, 180, 190 |
| 11. | CPDCT035 | 4 | $(GA)_{17}$ | $52^+$ | 162, 163, 164, 165 |
| 12. | CPDCT038/1 | 2 | $(GA)_{25}$ | 62 | 156, 160 |
| 13. | CPDCT038/2 | 2 | $(GA)_{25}$ | 62 | 181, 189 |
| 14. | CPDCT038/3 | 3 | $(GA)_{25}$ | 62 | 274, 311, 329 |
| 15. | CPDCT039 | 3 | $(GA)_{15}$ | 62 | 154, 155, 156 |
| 16. | CPDCT040 | 3 | $(GA)_{24}$ | 62 | 144, 145, 146 |
| 17. | CPDCT043 | 3 | $(GA)_{21}$ | $48^+$ | 71, 76, 102 |
| 18. | CPDCT045 | 5 | $(GA)_{16}$ | 62 | 139, 144, 149, 159, 167 |
| 19. | CPDCT046 | 5 | $(GA)_{21}$ | 62 | 149, 150, 151, 152, 153 |

**Notes.**
$^+$—Optimized annealing temperature (°C).

**Table 2** Assessment of genetic diversity in studied populations of *P. ledebouriana* and *P. tenella*.

| ID | Species | Mean/SE | Na | Ne | uHe | Fst |
|---|---|---|---|---|---|---|
| 1-UR | *P. ledebouriana* ($n = 20$) | Mean | 4.158 | 2.427 | 0.560 | 0,060 |
| | | SE | 0.906 | 0.177 | 0.037 | |
| 2-KO | *P. ledebouriana* ($n = 20$) | Mean | 3.316 | 2.106 | 0.483 | 0,113 |
| | | SE | 0.991 | 0.158 | 0.045 | |
| 3-KA | *P. ledebouriana* ($n = 20$) | Mean | 3.316 | 2.108 | 0.461 | 0,057 |
| | | SE | 1.000 | 0.179 | 0.056 | |
| 4-UK | *P. tenella* ($n = 20$) | Mean | 4.632 | 2.868 | 0.622 | 0,066 |
| | | SE | 1.041 | 0.338 | 0.025 | |

**Notes.**
$n$, number of plants studied in the population; Na, number of alleles per locus; Ne, effective number of alleles; uHe, Unbiased Expected Heterozygosity; Fst, fixation index.

73% within populations and 27% between populations. The evaluation of the partitioning of the genetic variation across four populations (including 4-UK *P. tenella*) resulted in a decrease in the level of variation within populations (63%) and an increase in variation between populations (37%).

The *t*-test was applied to test associations between SSR markers and plant height in samples of both species. The results showed that nine SSR loci were statistically associated with plant height (Table 4).

**Table 3 Assessment of genetic diversity of SSR loci in the analysis of *P. ledebouriana* and *P. tenella* populations.**

| # | Locus | $H_O$/ $H_E$ | FIS | PIC | Rp | MI |
|---|-------|-----------|-----|-----|-----|-----|
| 1 | CPDCT005 | 0.719/0.280 | 0.252 | 0.251 | 0.282 | 0.279 |
| 2 | CPDCT007 | 0.711/0.288 | 0.400 | 0.283 | 0.307 | 0.304 |
| 3 | CPDCT008 | 0.793/0.206 | 0.288 | 0.194 | 0.207 | 0.205 |
| 4 | CPDCT012 | 0.584/0.415 | 0.342 | 0.345 | 0.432 | 0.427 |
| 5 | CPDCT015 | 0.703/0.296 | 0.104 | 0.314 | 0.333 | 0.329 |
| 6 | CPDCT016 | 0.570/0.429 | 0.122 | 0.345 | 0.432 | 0.427 |
| 7 | CPDCT022 | 0.856/0.143 | 0.124 | 0.158 | 0.166 | 0.164 |
| 8 | CPDCT023 | 0.502/0.497 | 0.694 | 0.403 | 0.500 | 0.494 |
| 9 | CPDCT025 | 0.390/0.609 | 0.407 | 0.524 | 0.612 | 0.605 |
| 10 | CPDCT027 | 0.387/0.612 | 0.563 | 0.535 | 0.616 | 0.609 |
| 11 | CPDCT035 | 0.502/0.497 | 0.491 | 0.461 | 0.500 | 0.494 |
| 12 | CPDCT038/1 | 0.635/0.364 | 0.327 | 0.296 | 0.366 | 0.362 |
| 13 | CPDCT038/2 | 0.860/0.139 | 0.243 | 0.129 | 0.140 | 0.138 |
| 14 | CPDCT038/3 | 0.445/0.554 | 0.179 | 0.511 | 0.590 | 0.582 |
| 15 | CPDCT039 | 0.432/0.567 | 0.086 | 0.468 | 0.570 | 0.563 |
| 16 | CPDCT040 | 0.434/0.565 | 0.317 | 0.467 | 0.568 | 0.561 |
| 17 | CPDCT043 | 0.485/0.514 | 0.359 | 0.453 | 0.517 | 0.511 |
| 18 | CPDCT045 | 0.398/0.601 | 0.120 | 0.529 | 0.615 | 0.607 |
| 19 | CPDCT046 | 0.309/0.690 | 0.135 | 0.631 | 0.695 | 0.686 |
| | Mean | 0.586/0.413 | 0.308 | 0.384 | 0.445 | 0.693 |

**Notes.**

$H_O$, Observed heterozygosity; $H_E$, Expected heterozygosity; FIS, Fixation index; Rp, Resolving power; MI, Marker index.

## Population structure in samples of four studied populations using SSR markers

All *P. ledebouriana* and *P. tenella* samples were analyzed for population structure using the package STRUCTURE based on genotyping data from 19 polymorphic SSR markers. The structure was assessed using results from $K = 2$ to $K = 10$. The assessment of K plots suggested that starting from $K = 3$ and $K = 4$ (Fig. 3), plants from population 4-UK were separated from three populations of *P. ledebouriana*. Interestingly, the population 3-KA, growing at the highest elevation and characterized by the lowest level of genetic diversity within four studied populations (Table 2), was drifting apart from other groups of plants in step $K = 2$. In the analysis of principal coordinates (PCoA), it was determined that the first and second principal coordinates described 49.05% and 41.16% of the variability, respectively (Fig. 4). PC1 effectively separated 3-KA from the other three populations. At the same time, PC2 allowed for the differentiation of 4-UK from 1-UR and 2-KO. In addition, the UPGMA dendrogram was built based on the genotyping results for samples in four populations. The results suggested that population 4-UK formed a distinct cluster, and only one sample from that population (4-UK_07) was positioned close to the cluster with the domination of samples from population 3-KA. Likewise, in PCoA analysis, the UPGMA dendrogram distinguished 3-KA from 1-UR and 2-KO. At the same time, the latter populations had a mix of samples in several clades (Fig. 5).

**Table 4 Results of association assessment between SSR loci and plant height in populations of *P. ledebouriana* and *P. tenella* using *t*-tests.**

| # | Marker ID | Allele | *n* | Average PH | *p*-value |
|---|---|---|---|---|---|
| 1 | CPDCT005 | C | 67 | 182.8 | 0.000179*** |
|  |  | B | 11 | 144.6 |  |
| 2 | CPDCT007 | B | 66 | 183.5 | 0.000125*** |
|  |  | A | 9 | 140.2 |  |
|  |  | C | 4 | 150.7 |  |
| 3 | CPDCT025 | B | 33 | 166.8 | 9.77e−05*** |
|  |  | C | 21 | 180.5 |  |
|  |  | A | 8 | 151.5 |  |
| 4 | CPDCT027 | C | 27 | 179.7 | 9.78e−05*** |
|  |  | A | 25 | 197.6 |  |
|  |  | B | 14 | 172.2 |  |
| 5 | CPDCT035 | A | 55 | 186 | 1.81e−08*** |
|  |  | D | 10 | 136.5 |  |
|  |  | C | 8 | 147.8 |  |
|  |  | B | 7 | 195.7 |  |
| 6 | CPDCT038_3 | C | 44 | 170.8 | 0.00883** |
|  |  | A | 26 | 192.3 |  |
|  |  | B | 7 | 156.7 |  |
| 7 | CPDCT040 | C | 41 | 188.8 | 0.000729*** |
|  |  | B | 33 | 166.1 |  |
|  |  | A | 6 | 153.6 |  |
| 8 | CPDCT043 | A | 52 | 166.5 | 1.39e−05*** |
|  |  | B | 18 | 204.1 |  |
|  |  | C | 10 | 181.1 |  |
|  |  | D | 32 | 193.7 |  |
| 9 | CPDCT046 | A | 29 | 172.3 | 0.000134*** |
|  |  | B | 10 | 157 |  |
|  |  | E | 5 | 143.8 |  |
|  |  | C | 4 | 165.2 |  |

**Notes.**

*P*—values are provided with significance level indicated by the asterisks.

*$P < 0.05$.

**$P < 0.01$.

***$P < 0.001$.

# DISCUSSION

## The phylogenetic relationship between *P. tenella* and *P. ledebouriana*

The phylogeny of the genus *Prunus* is complex. Previous reports provided contradictory results on the relationships of plum species (*Badenes & Parfitt, 1995*). Nevertheless, it has been well established that *P. tenella* belongs to the section *Amygdalopsis* within the subgenus *Amygdalus* (*Avdeev, 2016*). The complexity of taxonomy in species within this section could be more precise, as there are questions about the relationship among different taxa. These questions include the taxonomic relationship between *P. tenella*

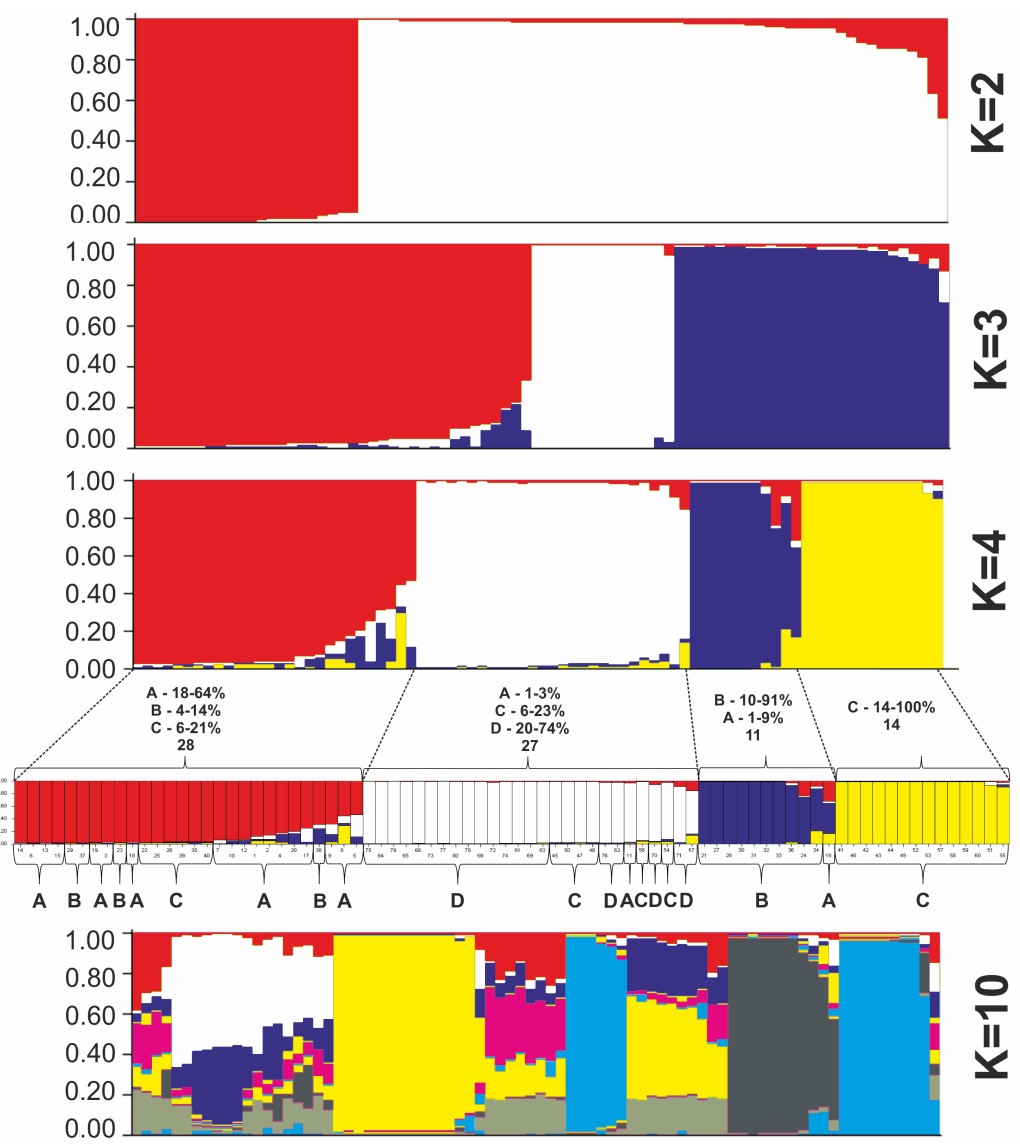

**Figure 3** Bayesian clustering of 60 *P. ledebouriana* (A—Urdzhar, B—Kokpekty, C—Katon-Karagay) and 20 *P. tenella* (D—Ulansky) plants at K = 2, K = 3, K = 4, and K = 10 step.

and *P. ledebouriana*, where the former is widespread in the Eurasian continent, and the latter is limited to mountainous populations of East Kazakhstan, particularly in the Altai mountains (*Zhukovsky, 1971*; *Dzhangaliev, Salova & Turekhanova, 2003*; *Myrzagalieva et al., 2015*; *Myrzagalieva & Orazov, 2018*). Several reports proposed that these two species have minor differences and could be considered one species (*Sokolov, Svyazeva & Kubli, 1980*; *Qiu et al., 2012*).

Conversely, publications suggest that *P. ledebouriana* and *P. tenella* have sufficient morphological differences to separate them into two distinct species (*Zaurov et al., 2015*; *Orazov et al., 2020*). Plant height is one of the most prominent traits used to differentiate

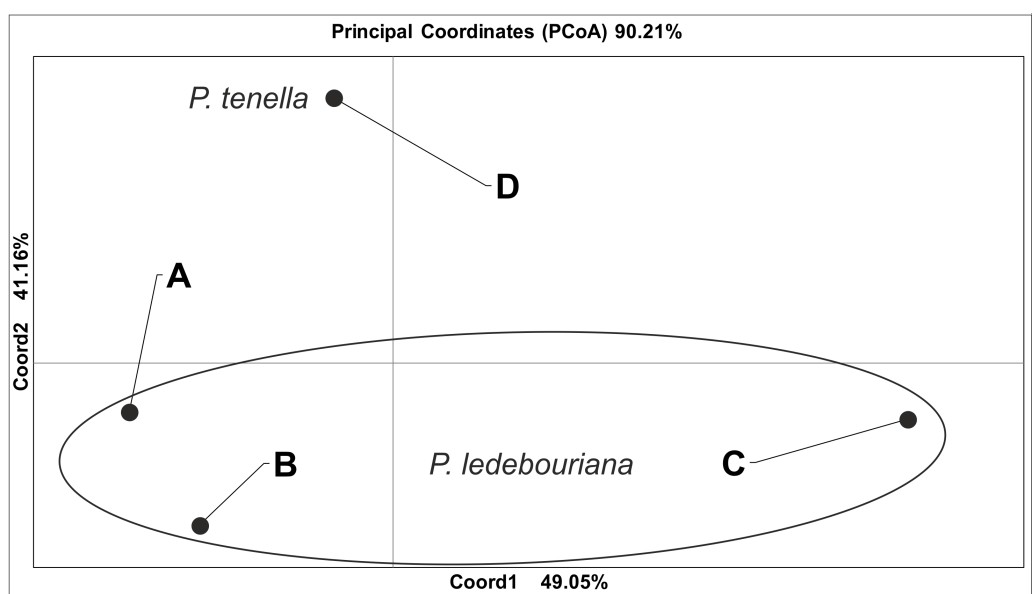

**Figure 4 Principal coordinate analysis (PCoA) for populations of *P. ledebouriana* (A—Urdzhar, B—Kokpekty, C—Katon-Karagay) and *P. tenella* (D—Ulansky) using polymorphic SSR loci.**

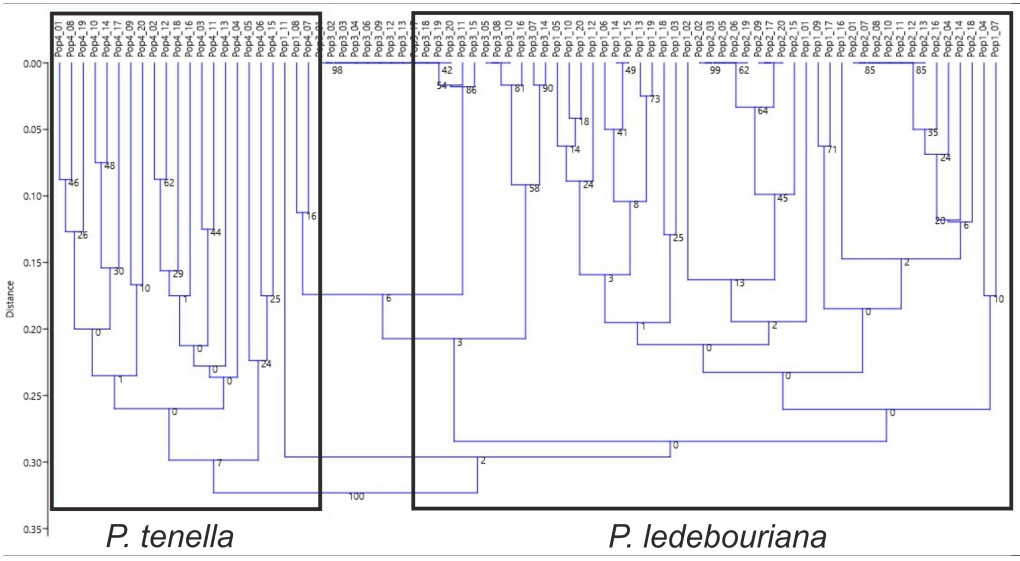

**Figure 5 Unweighted pair group method with arithmetic mean (UPGMA) dendrogram of *P. ledebouriana* (Pop1—Urdzhar, Pop2—Kokpekty, Pop3—Katon-Karagay) and *P. tenella* (Pop4—Ulansky) using polymorphic SSR loci.**

these species (*Mushegyan, 1962*; *Vintereoller, 1976*). In this work, all plants from the three *P. ledebouriana* populations and one population *of P. tenella* were measured for plant height. The results clearly showed that samples from the two species have a distinct separation based on this trait ($P < 0.0001$). Similar results for different morphological characteristics

confirm the difference among other representatives of the genus (*Devi, Singh & Thakur, 2018*). Hence, evaluating plant height can be a reliable way of distinguishing between two closely genetically related species within the section *Amygdalopsis*. The performance of this trait in plants is reliably related to the elevation above sea level (a.s.l.), as the lowest altitude for *P. ledebouriana* samples was higher than 1.7 m a.s.l. and the highest altitude for *P. tenella* samples was lower than 1.5 m a.s.l. (Table S1). We assessed samples from two species using 19 polymorphic SSR loci to confirm the conclusion based on the plant height study. The application of SSR markers resulted in a constructed UPGMA phylogenetic tree that separated 20 samples of *P. tenella* (4-UK) from 60 samples of *P. ledebouriana* (1-UR, 2-KO, 3-KA) (Fig. 4). This result was also supported by the PCoA plot, where PC2 (41.2%) split 4-UK samples from 1-UR and 2-KO.

In contrast, PC1 (49.1%) separated 3-KA from the remaining three populations (Fig. 3). Interestingly, the clusterization in Fig. 4 suggests that, generally, there is a low level of admixture between populations, which supports the model of "isolation by distance" (*Meirmans, 2012*). The genetic heterozygosity index (Nei's index) assessment showed that the highest genetic diversity was registered in the population of *P. tenella* (0.606). In contrast, the lowest index was recorded in population 3-KA (0.449), representing the area with the highest sampling elevation. High altitude is a sufficiently solid environmental factor that negatively influences genetic variation in *P. ledebouriana*. Nevertheless, the separation of 3-KA from 1-UR and 2-KO supported a more significant genetic variation within the species. The high genetic differentiation between mountain and steppe populations is most likely due to the factor of the steppe zone and the presence of anthropogenic pressure on the *P. tenella* (4-UK) population. In turn, mountain populations are distinguished by mountain isolation of ridges. The analysis of samples by DNA genotyping using SSR markers suggested that five loci were characterized as markers with the highest polymorphism level (Table 3). These five SSR loci could also be recommended for the discrimination of *Prunus* species in other studies. In addition, a *t*-test was applied to test the association of 19 polymorphic SSR loci with plant height (Table 4). It was concluded that nine out of 19 SSR loci were significantly associated with plant height. This result may not be a direct reflection of associations between SSRs and plant height but an indication of the genetic differences between *P. ledebouriana* and *P. tenella,* as these two species have significantly differed in plant height (Fig. 1). Therefore, these nine SSR loci can be efficiently used in further studies of discrimination between *P. ledebouriana* and *P. tenella*. The assessment of the population structure using the STRUCTURE package suggested that populations of two species started separating at steps $K = 3$ and $K = 4$, which is another indication that *P. ledebouriana* and *P. tenella* are two different species. The evaluation of samples in four clusters at $K = 4$ (Fig. 3) showed little admixture level, supporting the model of isolation by distance with a limited gene flow among the populations. Mantel tests revealed a positive correlation between geographic and genetic distance among populations ($r = 0.4387$), demonstrating consistency with the isolation-by-distance model.

## CONCLUSION

Discriminating the endemic species *P. ledebouriana* from wild almond *P. tenella* has been a poorly studied issue for the genus *Prunus*. In this work, two different approaches were conducted to analyze the genetic relationship between these two closely related species. In the first approach, the detailed analysis of plant height from one population of *P. tenella* and three populations of *P. ledebouriana* allowed a significant separation ($P < 0.0001$) between the two species. In the second approach, the samples of these four populations were genotyped using 19 polymorphic SSR loci. The NJ phylogenetic tree and PCoA plot also showed a significant separation of two species on two groups of clusters. Also, the UPGMA dendrogram and PCoA plot have demonstrated that within *P. ledebouriana*, the population 3-KA sharply differed from populations 1-UR and 2-KO, supporting a high diversity level within the species. The assessment of the connections between SSR loci and plant height showed that nine out of 19 loci were associated with the studied morphological trait, suggesting that these loci can be efficiently used in DNA discrimination of two species. The population structure analysis suggested that samples in two species were separated starting from step $K = 3$. The assessment of plants in clusters at steps $K = 3$ and $K = 4$ suggested a limited admixture level between populations, supporting the model of isolation by distance. Thus, the analysis of plant height and application of SSR markers were successfully used to discriminate *P. tenella* and *P. ledebouriana* and study the genetic diversity and population structure of the endemic species *P. ledebouriana*. A clear distinction between similar plants from different populations makes it possible to delineate the boundary of mutual replacement of the *P. tenella* and *P. ledebouriana* species. It will be possible to accurately separate precious endemic populations of *P. ledebouriana* from the simple *P. tenella*, and then clarify the taxonomy of the genus and organize conservation measures in the mountainous zones of Eastern Kazakhstan. The obtained data on SSR will allow further research at a higher level. It is proposed to use markers such as ITS, s6pdh, trnL-trnF, and trnS-trnG to compare several species of shrub almonds from Central Asia and use whole-genome methods.

## ACKNOWLEDGEMENTS

The authors express their gratitude to the head of the Department of Science, Information, and Monitoring of the State National Natural Park, Tarbagatai Alemseitova Janylkan Kabikyzy, for organizing field scientific expeditions in the territory of the National Park.

### Funding

This research has been funded by the Science Committee of the Ministry of Science and Higher Education of the Republic of Kazakhstan (Grant No. AP19177914 ''Study of genetic polymorphism of endemic representatives of Prunus subgenus. Amygdalus (L.) Focke on different mountain ranges of Kazakhstan'' 2023–2025 at the Environmental Research

Laboratory "NatureLab" of Astana International University of the Ministry of Science and Higher Education of the Republic of Kazakhstan of the Republic of Kazakhstan. AP05131621 "Information system for molecular genetics and botanical documentation of the wild flora of Kazakhstan" for 2018–2020 in the Laboratory of Molecular Genetics, Institute of Plant Biology and Biotechnology of the Ministry of Science and Higher Education of the Republic of Kazakhstan of the Republic of Kazakhstan. The funders had no role in study design, data collection and analysis, decision to publish, or preparation of the manuscript.

## Grant Disclosures

The following grant information was disclosed by the authors:
Science Committee of the Ministry of Science and Higher Education of the Republic of Kazakhstan: AP19177914.
"Study of genetic polymorphism of endemic representatives of Prunus subgenus. Amygdalus (L.) Focke on different mountain ranges of Kazakhstan" 2023–2025 at the Environmental Research Laboratory "NatureLab" of Astana International University of the Ministry of Science and Higher Education of the Republic of Kazakhstan of the Republic of Kazakhstan.
"Information system for molecular genetics and botanical documentation of the wild flora of Kazakhstan" for 2018–2020 in the Laboratory of Molecular Genetics, Institute of Plant Biology and Biotechnology of the Ministry of Science and Higher Education of the Republic of Kazakhstan of the Republic of Kazakhstan: AP05131621.

## Competing Interests

The authors declare there are no competing interests.

## Author Contributions

- Aidyn Orazov conceived and designed the experiments, performed the experiments, analyzed the data, prepared figures and/or tables, getting the main results, and approved the final draft.
- Moldir Yermagambetova performed the experiments, analyzed the data, prepared figures and/or tables, getting the main results, and approved the final draft.
- Anar Myrzagaliyeva analyzed the data, authored or reviewed drafts of the article, scientific advice, and approved the final draft.
- Nashtay Mukhitdinov conceived and designed the experiments, authored or reviewed drafts of the article, scientific advice, and approved the final draft.
- Shynar Tustubayeva analyzed the data, prepared figures and/or tables, getting the main results, and approved the final draft.
- Yerlan Turuspekov conceived and designed the experiments, analyzed the data, authored or reviewed drafts of the article, getting the main results, and approved the final draft.
- Shyryn Almerekova conceived and designed the experiments, performed the experiments, analyzed the data, prepared figures and/or tables, getting the main results, and approved the final draft.

## Data Availability

Additional genetic and general information about populations are available in the Supplemental Files.

## Supplemental Information

Supplemental information for this article can be found online at http://dx.doi.org/10.7717/peerj.16735#supplemental-information.

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
