# Peer review of "Plant height variation and genetic diversity between Prunus ledebouriana (Schlecht.) YY Yao and Prunus tenella Batsch based on using SSR markers in East Kazakhstan"

_PeerJ, doi:10.7717/peerj.16735_

## Round 0.1 · original submission · Major Revisions

Dear Dr. Orazov,

Thank you for your submission to PeerJ.

Based on review reports and my own assessment, your article - Plant height variation and genetic diversity of Prunus ledebouriana (Schlecht.) Y.Y.Yao based on using SSR markers in East Kazakhstan - requires a number of Major Revisions.

The reviewers have underlined several major and minor shortcomings which you need to take into account and thoroughly revise the manuscript to make it suitable for publication. It is worth mentioning that your revised paper will be re-evaluated to ensure that you have carefully addressed all the queries, comments and suggestions raised by the reviewers. You are advised to maintain coherence between different sections of the manuscript to attract wide readership. Specifically, carefully address the reviewers comments in relation to gel image, some essential marker parameters, introduction, materials and discussion sections, and place utmost emphasis on improving the English language.

Hope to receive the revised draft soon.

**Language Note:** The Academic Editor has identified that the English language must be improved. PeerJ can provide language editing services - please contact us at copyediting@peerj.com for pricing (be sure to provide your manuscript number and title). Alternatively, you should make your own arrangements to improve the language quality and provide details in your response letter. – PeerJ Staff

·

Basic reporting

The abstract is confusing and significantly laid, and it does not accurately reflect the importance of the
works. Precisely a result, it should be necessary to rewrite carefully.

Experimental design

No comment

Validity of the findings

No comment

Additional comments

CONCLUSION and DISCUSSION: This section is not clear. Rewrite
Kindly include the SSR gel Image
Kindly replace the plant height Figure (BOX Plot model) with the column fig.

Reviewer 2 ·

Basic reporting

I have carefully reviewed the MS entitled “Plant height variation and genetic diversity of Prunus ledebouriana (Schlecht.) Y.Y.Yao based on using SSR markers in East Kazakhstan”. But regret to inform that the MS needs more upgradation. The results and discussion part is very weak which should be written scientifically with more interpretation. The comments below are meant to help the quality of the manuscript:
1. The introduction part needs revision, please focus on main idea and literatures, clearly write about its application. Uses of the plant should be mentioned. Why is it important to study about this plant?
2. How can the plant be both cultivated and wild? Elaborate.
3. Novelty and gap of the work should be mentioned.
4. Keywords missing.
5. The genotypes in a particular population along with its code should be presented in the MS.
6. Why 50 ng of template DNA was used for PCR reaction when the DNA was normalized to 20 ng/microlitre? Write the volume of the materials used for PCR mix apart from the concentrations.
7. The meaning of the sentences 154-156 is not clear. Whether the 19 SSR markers or 17 were found to be polymorphic.
8. The results written are very limited. The discussion part is the repetition of the results and lacks interpretation of the results.
9. What is “reference” in line number 225?
10. S. No. 17 in table 2 has missing row.
11. Authors must include PCR data (gel image) for at least two best primers used in this study.
12. Why was the efficiency of the primers not checked? The parameters like MI, Rp and polymorphic% should also be calculated along with PIC. Refer to the recent papers for clarity:
 Munda et al., 2022. Evaluation of genetic diversity based on microsatellites andphytochemical markers of core collection of Cymbopogon winterianusJowitt.germplasm. Plants 11 (4), 528.
 Lal et al., 2022. Molecular genetic diversity analysis using SSR marker amongst highsolasodine content lines of Solanum khasianum C.B. Clarke, an industriallyimportant plant. Industrial Crops & Products 184 (2022) 115073.

Experimental design

NA

Validity of the findings

NA

Additional comments

NA

Reviewer 3 ·

Basic reporting

Orazov et al., in this manuscript, performed various methods to measure the morphological and genetic variations within P. ledebouriana populations as well as between P. ledebouriana and P. tenella populations. Their results indicate that P. ledebouriana and P. tenell more likely belong to different species.

The main concerns list as follows:

1. The English language should be improved to ensure that an international audience can clearly understand your text. Some examples where the language could be improved include lines 24-25, line 43-44, line 53-55, and line 144-145– the current phrasing has obvious grammatical errors and makes comprehension difficult. I suggest you have a colleague who is proficient in English and familiar with the subject matter review your manuscript or contact a professional editing service.

2. The title is not appropriate because in the manuscript you also did a bunch of work to analyze the P. tenella rather than only focusing on P. ledebouriana.

3. You should spell out the full name of “USSR”, “SSR” and “PIC” which are in line 11, line 26, and line 161, respectively.

4. The quality of Figs and tables need to be improved. For Fig.1, you should add proper p-value information. For example, P<0.0001 can be marked using 3 stars. Moreover, in Fig.1, different populations should be showed using different colors of bars (rather than only green). For Fig.3, you would better use coord1 (49.05%) and coord1 (41.16%) to avoid confusing, and you should also note that you can not use commas to replace decimal points. Furthermore, for Fig.3, you would better use different color dots to represent the different populations (rather than only blue). For table 3, Fst should change into FIS. For Table 4, you can not use commas to replace decimal points, and what does “number of four samples in populations” mean? For Table 5, please show the meaning of 3 stars.

5. Some small errors need to be corrected. 1), in line 31-33, a bracket is missing at the back of the “WFO)”. 2), in line 33, what does GBIF mean? 3), in line 111, 10ml reactions should change into 10ul reactions, right? 4), in line 159-160, Table 4 should change into Table 2. 5), In line 144-148, all SDs you showed (0.06, 0.03, 0.03, and 1.04) are different from the values you showed in Table 3. Moreover, for 1-UR, the plant height is 2 rather than 1.78 according to Table 3. 6), in line 163, “above 0.3” should change into “0.3 to 0.5”. 7) in line 163, “bellow 0.2” should change into “bellow 0.3”. 8), in line 165, I can not find the 0.456, and it should be 0.622? 9), in line 193, “former” should be corrected to “latter”. 10), in line 202-203, please double check to confirm that the positions of “former” and “latter” is right. 11) in line 227-228, I did not find 0,606 and 0.449 in anywhere. 12), you did not mention the supplementary table in the manuscript.

Experimental design

6. You must provide clear associations between the methods you described in Materials & Methods, such as Shannon diversity index, FIS (inbreeding coefficient) and AMOVA, with the various values you showed in the results such as PIC, Nei's index, HO/HE, and uHe. In other words, you should explain your methods in more detail, or at least unifying your statement.

7. In lines 167-171, you should be more specific about how to get the values of 73%, 27%, 63% and 37%.

8. In line 149, please clarify which populations you compared to get the P-value of 2.3e-15.

9. In line 179, 180, 183, you mentioned K=2, K=3, K=4 and K=10, so you should provide these results rather than providing only results of K=4 (Fig. 2)

Validity of the findings

10. It would be better if you can appropriately broaden the meaning of your research in the conclusion. For example, what positive impact does your research have on future related studies? Or whether your study provides help for species conservation?

Annotated reviews are not available for download in order to protect the identity of reviewers who chose to remain anonymous.

·

Basic reporting

In the manuscript entitled “Plant height variation and genetic diversity of Prunus ledebouriana (Schlecht.) Y.Y.Yao based on using SSR markers in East Kazakhstan” by Orazov et al. The authors have conducted genetic diversity analysis in Prunus ledebouriana ssp. by using SSR based markers for variability in plant height. The authors used 17 polymorphic (SSR) markers to assess molecular genetic diversity and relatedness. The informative marker combinations revealed a total of 14 alleles at 19 loci. The authors have followed a systematic approach for delineating the diversity analysis using the molecular approach in the test population. However, there are certain suggestions and queries for further improvement of the manuscript.

Experimental design

no comment

Validity of the findings

no comment

Additional comments

1. Introduction part is too long. It should be specific to the work done in present investigation.
2. The mean polymorphic information content (PIC) value should be mentioned in the results.
3. In line No. 161, rather than using the word ‘groups’, a subpopulation term would have been appropriate when referring to the population being analyzed in the present study.
4. In line No. 225 reference is missing.
5. What could be the probable reason for the relatively high population differentiation was amongst the test populations as indicated by the mean fixation index (Fst) value? Similar should be mentioned in the discussion part.
6. In line no. 71 the full form of ISSR ‘simples’ should be replaced by ‘simple’.
7. In line No. 111 “report of Mnejja with co-authors (2005)” could be written as “reported by Mnejja et al., 2005”.
8. The authors have postulated that higher altitude tends to negatively affect the genetic diversity of test species. Does the altitude negatively affect the plant height and similar relationship if found any could have been elaborated on in the discussion section.
9. The authors should briefly elaborate on the clustering pattern revealed and the relationship between geographic origin and genetic diversity in the context to present study.
10. Table No. 1 could be moved to the supplementary material.
11. The quality and appearance of Figure no. 4 needs to be enhanced.
12. Under the discussion, a part should also discuss diversity with regard to earlier reports in height variation among the Prunus sp.

---

## Round 0.2 · accepted · Accept

I can confirm that the revision is Acceptable in my view

Reviewer 3 ·

Basic reporting

The manuscript has greatly improved, and I suggest accept.

Experimental design

NA

Validity of the findings

NA

Additional comments

NA